# Organoid Models for Cancer Research—From Bed to Bench Side and Back

**DOI:** 10.3390/cancers13194812

**Published:** 2021-09-26

**Authors:** Carolin Kastner, Anne Hendricks, Hanna Deinlein, Mohammed Hankir, Christoph-Thomas Germer, Stefanie Schmidt, Armin Wiegering

**Affiliations:** 1Department of General, Visceral, Transplantation, Vascular and Pediatric Surgery, University Hospital, University of Wuerzburg, Oberduerrbacherstr. 6, 97080 Wuerzburg, Germany; kastner_c@ukw.de (C.K.); Hendricks_A@ukw.de (A.H.); hanna.deinlein@stud-mail.uni-wuerzburg.de (H.D.); hankir_m@ukw.de (M.H.); germer_c@ukw.de (C.-T.G.); stefanie.schmidt2@biozentrum.uni-wuerzburg.de (S.S.); 2Department of Biochemistry and Molecular Biology, University of Wuerzburg, Am Hubland, 97074 Würzburg, Germany; 3Comprehensive Cancer Centre Mainfranken, University of Wuerzburg Medical Centre, Josef-Schneiderstr. 2, 97080 Wuerzburg, Germany

**Keywords:** cancer, tumor disease, organoid, patient-derived organoid (PDOs), patient-derived tumor organoid (PDTO)

## Abstract

**Simple Summary:**

Despite significant strides in multimodal therapy, cancers still rank within the first three causes of death especially in industrial nations. A lack of individualized approaches and accurate preclinical models are amongst the major barriers that limit the development of novel therapeutic options and drugs. Recently, the 3D culture system of organoids was developed which stably retains the genetic and phenotypic characteristics of the original tissue, healthy as well as diseased. In this review, we summarize current data and evidence on the relevance and reliability of such organoid culture systems in cancer research, focusing on their role in drug investigations (in a personalized manner).

**Abstract:**

Organoids are a new 3D ex vivo culture system that have been applied in various fields of biomedical research. First isolated from the murine small intestine, they have since been established from a wide range of organs and tissues, both in healthy and diseased states. Organoids genetically, functionally and phenotypically retain the characteristics of their tissue of origin even after multiple passages, making them a valuable tool in studying various physiologic and pathophysiologic processes. The finding that organoids can also be established from tumor tissue or can be engineered to recapitulate tumor tissue has dramatically increased their use in cancer research. In this review, we discuss the potential of organoids to close the gap between preclinical in vitro and in vivo models as well as clinical trials in cancer research focusing on drug investigation and development.

## 1. Introduction

Despite major advances in all aspects of multimodal therapy, cancer remains one of the leading causes of death worldwide, accounting for almost 10 million deaths in 2020 [1,2]. Pioneering work on the molecular mechanisms underlying tumor growth and in the development of anticancer drugs was largely performed on immortalized cancer cell lines [3,4]. However, classic 2D cell cultures suffer from several drawbacks, making them an unideal model to study cancer disease and to test drugs, especially in a personalized manner. Additionally, immortalized cancer cell lines undergo fundamental changes in their molecular characteristics while being established from patient or animal tissue and further alterations in their genetic/mutational landscape can occur after long-term passage in cell culture. Moreover, the heterogeneity of the initial tumor tissue as well as cellular plasticity and hierarchy cannot be recapitulated [5,6,7]. This limiting factor not only applies to tumor tissue from different patients, but also for tumor tissue from the same patient.

There are several other limitations when trying to extrapolate results from model systems such as 2D cell cultures to the human in vivo situation [8]. This might explain why only 3.4% of cancer-targeting drugs are approved for clinical use after passing the clinical trial process [9,10,11]. Furthermore, the idea of individualized therapy by deciphering a specific patient’s mutational landscape has stalled in recent years. Even if gene sequencing becomes widespread, only a few relevant and targetable mutations have been identified [12].

To overcome these deficiencies, extensive efforts have been made to create ex vivo model systems that better mimic the human in vivo situation. In 2009, the group of H. Clevers set a milestone when they described the establishment of a murine intestinal crypt culture system called organoids [13]. We now know that starting from a (tissue-specific) stem/progenitor cell, organoids are self-organizing organotypic structures that at least partially show cellular heterogeneity, basic functional and structural aspects of the tissue of origin [14]. Therefore, based on their properties, organoids are thought to fill the gap between “classic” cell lines and in vivo models.

## 2. Origin and Establishment of Organoid Models

Especially in the context of cancer research, a variety of 3D culture systems has been developed to meet deficiencies of 2D cell culture during recent decades [15,16]. In addition to different concepts of spheres or spheroids, the 3D culture system of organoids arose and have been widely spread in recent years.

These tissue culture models are originally grown from (tissue-specific) stem/progenitor cells. Thus, two major types of organoids can be categorized. The first type derives from embryonic pluripotent stem cells or induced pluripotent stem cells (iPSCs) obtained by reprogramming of somatic cells [17,18]. The second type derives from adult stem cells (AdSCs) of the desired tissue [8].

(i)PSCs can differentiate into organoids resembling organs from all three germ layers depending on the differentiation signals provided [18,19,20]. Due to the necessity of simulating physiologic differentiation processes for their generation, organoids from (i)PSCs represent an excellent model system to study developmental processes and genetic disease [21,22,23,24,25]. Additionally, (i)PSCs are an important source of organoids from poorly accessible organs such as a healthy brain or optic cup/retina [8,14].

Compared to the sophisticated processes of reprogramming and differentiation to set up (i)PSC-derived organoids, AdSCs, for the establishment of corresponding organoids, can be obtained by biopsy or from surgically resected specimens of normal or diseased epithelial tissue of the organ of interest (see Figure 1). One of the main requirements to isolate, culture and grow AdSCs into tissue resembling organoids is the knowledge of tissue-specific properties and potential markers for identification as well as knowledge of the tissue-specific culture conditions [8]. The resulting organoids have been shown to be genetically and epigenetically as well as phenotypically stable even during long-term passaging [26,27,28,29]. However, compared to (i)PSC-derived organoids, they are less complex in structure and predominantly limited to the epithelial cell type of origin [7,8,30]. Advances in culture methods of recent years such as air–liquid interface (ALI) or microfluidic 3D culture systems facilitate an enhanced organoid culture comprising components of (tumor) microenvironment in addition to the tissue specific epithelial organoids [31,32,33,34].

As these organoids retain many features of the epithelium of origin, such as architecture, composition in terms of differentiated cell types and functional aspects, they are regarded as an ideal model to study tumor disease more closely to “reality” ex vivo [35].

## 3. Organoids as a Valuable Model for the Investigation of Tumor Development and Progression

When studying tumor biology, one of the fundamental aspects is to identify driver mutations responsible for the transformation from normal to malign tissue. As a prime example, for colorectal cancer (CRC), it has been shown that the accumulation of different genetic alterations over time can fuel development and progression of malignant tumors [36,37]. For a deeper understanding of different tumor entities, reliable recapitulation and further investigation of genetic and epigenetic changes responsible for invasive growth or metastases is essential, but is not sufficient, in 2D cell cultures. Organoids from healthy epithelium (or (i)PSCs) that are genetically engineered, e.g., by usage of CRISPR/Cas9 or overexpression constructs, offer an optimized model to study the impact of distinct mutations in tumorigenesis [38,39,40,41,42,43,44,45,46,47,48,49].

To investigate the multihit oncogenesis theory of CRC, Matano et al. and Drost et al. independently introduced mutations in genes commonly altered in CRC such as APC, SMAD4, TP53, KRAS or PI_3_KCA [39,46]. In doing so, they modelled the oncogenic transformation of these organoids and highlighted a correlation between specific mutations and independency from distinct growth factors of the stem cell niche as well as the degree of invasiveness.

Editing of human colon organoids has also been used to gain better insight into less common premalignant lesions and CRC subtypes such as serrated adenoma [42,44]. Fessler et al. demonstrated that active TGFß signaling can already be found in early sessile serrated adenoma and plays a relevant role in driving them towards a mesenchymal, poor-prognosis CMS4 tumor subtype [44]. In addition, Kawasaki et al. used the CRISPR/Cas9 system in terms of chromosomal rearrangement of R-spondin genes, knock-in of BRAF mutations and GREM1 overexpression to model traditional serrated adenomas, a rare type of colorectal polyps with unique histological features, which could be re-established after xenotransplantation into mice [48].

Edited organoids from healthy epithelial tissue have not only been used to model carcinogenesis in CRC, but have also been applied in the setting of breast cancer and gastric cancer [45,47,49]. For example, shRNA-mediated knockdown of Tgfbr2 in murine, Cdh1−/−; Tp53−/− gastric organoids revealed the potential role of Tgfbr2 loss-of-function mutation in metastatic gastric cancer [45]. Additionally, by using the CRISPR/Cas9 approach, Seino et al. demonstrated that during tumorigenesis of pancreatic ductal adenocarcinoma (PDAC) mutations in PDAC driver genes such as KRAS, CDKN2A, SMAD4 and TP53 can lead to a situation of non-genetic Wnt independency [49]. In line with this, Liu et al. investigated factors associated with neoplastic transformation of premalignant esophageal lesions. In their work, CRIPR/Cas9 editing of human organoids elaborated the potential role of aberrant Wnt/β-catenin pathway activation for the development of neoplastic lesions in Barrett mucosa [43].

To establish engineered (i)PSC-derived organoids, gene editing can be performed directly in the initial (i)PSCs before starting a differentiation scheme [50]. This procedure, for example, has been used to establish and simulate a glioblastoma model [51]. Recently, Parisian et al. exploited the sequence of various differentiation stages during establishment of iPSC-derived organoids. They highlighted the benefit of iPSC-derived, engineered organoids for the investigation of the interplay of specific genetic alterations such as loss of function mutations in SMARCB1 and the differentiation state of the cell of origin. They demonstrated that type and strength of downstream effects of SMARCB1 shRNA-mediated knockdown vary depending on the cellular stage during differentiation scheme from iPSC to differentiated organoid. This gave deep insights into tumorigenesis of an early onset and highly aggressive, pediatric brain tumor (atypical teratoid rhabdoid tumors) [52]. This study emphasizes the benefit of organoid models in understanding developmental processes in tumorigenesis.

Finally, for profound comprehension of tumor progression and response to therapy, the knowledge about different tumoral cellular subtypes is essential. In this context, Tejero et al. used an inducible histone2B-GFP (iH2B-GFP) reporter to track cell division history in glioblastoma organoids. Their data supported the presence of a quiescent cell population with self-renewal capacity, high therapy resistance and mesenchymal gene signatures that could be a source of tumor progression and resistance under therapy [53].

## 4. Organoids as an Enhanced Model for (Personalized) Drug Screening

Establishment of AdSC-derived organoids has not only been described from healthy epithelial tissue, but also from patients’ tumor tissue—so called patient-derived tumor organoids (PD(T)Os). In 2011, Sato et al. first established an organoid culture system of benign and malign PDOs of the colon and esophagus [30]. These PDOs resemble the individual histopathological features, tumor stage, genetic landscape as well as cellular heterogeneity of the original tumor of the individual patient [54,55,56,57,58,59]. This applies not only for primary tumors but has also been shown for organoids derived from metastases or even from extremely rare circulating tumor cells (CTCs) [60,61,62]. In comparison to commonly used 2D cancer cell lines, tumor organoids allow long-term culturing while recapitulating and maintaining genetic, functional and phenotypic characteristics of initial tumor tissue without immortalization [63,64].

They offer a valuable opportunity to screen for individualized therapeutic options, to test treatment efficacy in advance to setting up patient’s therapeutic scheme and to analyze potential resistance mechanisms [65,66]. Therefore, PDOs may advance personalized medicine by predicting the clinical response of individual patients to antitumor agents.

Another advantage of PDOs is that they can easily be established from surgically resected tumor tissue or simple tissue biopsy, which implies less invasiveness for patients and enables the generation of PDOs from unresectable tumors as well [67]. For example, Tiriac et al. highlighted the feasibility of generating PDAC organoids from tissue samples gained by EUS-FNB (endosonographic ultrasound (EUS) fine needle biopsy (FNB)) [67]. Up to now, organoids from almost all endoderm-derived organs as well as their corresponding malign counterparts have been described. It is also possible to establish PDOs from different tumor subtypes of one organ such as adenocarcinoma and squamous cell carcinoma of the esophagus and the oropharynx, which mainly retain their tumor subtype characteristics in culture [68,69].

For broad range analyses covering different individual “subtypes” of a specific tumor entity, so called organoid biobanks have been initialized covering gastric, colorectal, breast, pancreatic, ovarian or bladder cancers (see Table 1) [49,63,64,67,70,71]. They comprise histologically and genetically characterized sets of tumor organoids, with or without matched wildtype organoids from a large number of patients.

## 5. Organoids as a Promising Tool for the Improvement of Therapy Efficiency Prediction

The relevance of organoids in a preclinical setting for therapeutic testing will partially be determined by the extent to which results can be extrapolated to patients´ therapy responses. A rapidly increasing number of studies are addressing the potential of PDOs as predictive markers for therapeutic response by correlating PDO treatment data to clinical treatment response.

In 2018, Vlachogiannis et al. showed for the first time that PDOs can serve as a valuable model to predict patients’ individual therapeutic responses to antitumoral drugs [90]. They compared the effect of anticancer agents (library of 55 drugs in Phase I to III clinical trials or in clinical use) in organoids and PDO-based orthotopic mouse tumor xenograft models from metastatic CRC, gastroesophageal and cholangiocarcinoma to the response of the original patients. They were able to show a 100% sensitivity, 93% specificity, and 88% and 100% positive and negative predictive values, respectively, when predicting patients´ therapeutic responses [90]. However, the analysis concerning correlation of clinical and PDO response was carried out retrospectively. In line with the findings of Vlachogiannis et al., Ooft et al. participated in a prospective, multicenter, observational study (TUMOROID study) to investigate the potential of PDO culture to predict the efficiency of several commonly used chemotherapeutic regimens (including infusional 5-fluorouracil (5-FU) or capecitabine, in combination with either oxaliplatin (FO) or irinotecan (FI), or irinotecan alone) in CRC patients [91]. In >80% of cases, PDO treatment assays precisely predicted response to irinotecan-containing treatment schemes (irinotecan single or combinatory treatment). However, it was not possible to predict the effect of 5-FU-oxaliplatin therapy, which was attributed to the lack of stroma and immune cells in the organoid model [91]. Ganesh et al. and Yao et al. demonstrated that the in vitro sensitivity of patient-derived rectal cancer organoids to chemotherapy and radiation correlated with clinical treatment results of corresponding patients [56,92]. The potential benefit of PDOs predicting clinical response to distinct therapeutic regimes has been shown in other entities besides CRC such as breast, gastric or pancreatic cancer as well as head and neck squamous cell carcinoma tumors (see Table 2) [63,70,75,93,94,95]. Recently, Beutel et al. set up a pancreatic cancer organoid biobank to study the feasibility of organoid application in prediction of therapeutic response for pancreatic cancer patients. Organoids were established from surgically resected or biopsy specimens, mainly from ductal adenocarcinoma but also from less frequent pancreatic cancer subtypes or liver metastases. Most patients were treatment-naïve (30/44) and in an advanced clinical stage (28/44). They tested five standard-of-care chemotherapeutics (gemcitabine, paclitaxel, irinotecan, 5-fluorouracil, and oxaliplatin), successfully obtained pharmacotyping of 28 organoid lines and implemented a score to check for beneficial drug combinations. For 16 patients, a correlation of preclinical results to performance was possible with a prediction accuracy of 91.1% in treatment-naïve patients for first-line regimen and of 80% in second-line therapy setting. For pretreated patients, the accuracy dropped to 40%, especially if more than one chemotherapeutic regime was administered before [96].

Up to now, most studies have been observational in nature. Only a small number of studies, such as that of Narasimhan et al. or the recently published one of Ooft et al., are of interventional design [97,98]. Ooft et al. performed a single-arm, single-center, prospective intervention trial to address whether optimal therapy for a patient can be chosen according to his/her organoids´ response in a drug screening assay [98]. This study was conducted on patients with metastatic, incurable CRC who passed first and second line chemotherapy but did not enter in last line/experimental therapy. Biopsies for tissue acquirement were carried out before and after the last part of current therapy. The tested agents comprised five FDA approved drugs and three drugs in an advanced phase of development (vistusertib, capivasertib, selumetinib, gefitinib, palbociclib, axitinib, gedatolisib and glasdegib). A drug screening was performed for 25 of the 61 initial patients; 19 patients were eligible to obtain study treatment and finally six patients received the allocated study medication based on a therapeutic screening of their organoids [98]. There was no significant and durable clinical response upon treatment with organoid-identified medication [98]. Explanations provided by the authors included poor blood–brain barrier penetrability of some of the used drugs, high dropout rate of patients due to clinical deterioration (both associated with advanced disease stage) and low efficiency of organoid establishment (31/54 (57%) successful culture establishment) [98].

To generate an overview of the current evidence on transferability of cancer organoid treatment results to the clinical setting, Verduin et al. and Wensink et al. conducted (systematic) reviews on the predictive value of tumor organoids [66,99]. Wensink et al. showed that 5 out of 17 studies included in their systematic review reported a statistically significant correlation and/or predictive value and 11/17 at least a trend for a correlation or predictive value [66]. Three studies did not find a relation between PDO and patients´ response and one study was unable to analyze this readout.

**Table 2 cancers-13-04812-t002:** Summation of some of the current evidence on transferability of organoid treatment results to clinical setting.

Tumor Type	Summation	Reference
Esophageal cancer	only small sample size; organoids derived from esophageal adenocarcinoma resemble the individual patients´ poor clinical response to classic chemotherapeutics such as 5-FU paclitaxel	[69,100]
Gastric cancer	correlation of treatment response of tumor organoids from primary tumor to therapeutic response of metastases for exemplary two patients	[70]
	ambiguous results in correlation of organoid treatment effects to patients´ clinical response	[101,102]
Colorectal cancer	Showing for the first time potential of organoids to predict clinical response; shown for metastatic CRC, gastroesophageal and cholangiocellular cancer	[90]
	APOLLO trial—first interventional trial; drug screening and next generation sequencing in organoids from peritoneal metastases of CRC; providing organoid-screening stratified therapy for 2 patients	[89]
	single-arm, single-center prospective intervention trial in metastatic CRC that missed to show feasibility of optimal therapy selection by organoid based drug screen	[98]
	ClinCare study—evaluating the predictive value of PDOs from 80 therapeutically naive locally advanced rectal cancer patients for patients´ clinical response to standard of care chemo(radio)therapy; sensitivity data of 68 organoids matched clinical outcome of the patients´, only 12 did not match	[103]
	retrospective correlation of treatment response of 7 rectal PDOs to corresponding patients´ clinical performance regarding 5-FU or FOLFOX treatment	[56]
Pancreatic cancer	prospective trial evaluating and correlating PDOs from primary or metastatic tissue as predictors of clinical drug response, mainly including ductal adenocarcinoma but also less frequent subtypes	[96]
	correlating treatment efficiency in PDOs to clinical results of the corresponding four patients for gemcitabine treatment demonstrating an overall correlation	[74]
	performing therapeutic profiling (pharmacotyping) in 66 PDOs for five mainly used chemotherapeutic agents in PDAC and retrospectively correlating patients´ outcome to their corresponding PDO performance demonstrating good correlation; longitudinal organoid sampling reflected patients´ individual clinical courses	[75]
	evaluating individual response of one patient with metastatic pancreatic cancer to PDO selected chemotherapy; PDO insensitivity to initial chemotherapeutic regime was represented in clinical setting as well as good response to PDO sensitive agents	[104]
Liver cancer	no study correlating PDO response to clinical performance	
Breast cancer	showing response correlation to tamoxifen of 12 PDOs from needle biopsy of metastatic breast cancer patients to the corresponding patient (their complete large PDO library (95 lines) could not be used for treatment response matching due to the establishment of organoids from surgically in sano resected tumor)	[63]
	drug identification for one patient by PDO drug screen	[105]
Ovarian cancer	showing statistically significant correlation in response of seven PDOs of five patients with high grade serous ovarian cancer to patients´ clinical history under carboplatin/paclitaxel therapy	[106]
HNSCC	matching PDO response to radiotherapy to the effects in the corresponding patients, good correlation	[95]
Glioblastom	evaluating glioblastoma organoid reaction to standard-of-care post-surgical treatment (temzolomide and radiation) to patients´ clinical performance with tendency to positive correlation; no prediction of treatment response by MGMT methylation status	[83]
Melanoma	establishing PDOs and corresponding immune-enhanced PDOs (iPDO) by usage of matching lymphnodes or WBC, treatment with different kinds of immunotherapeutic drugs (pembrolizumab, nivolumab, ipilimumab, dabrafenib/trametinib); positive correlation of iPDOs to patients´ performance in 85%; in addition longitudinal evaluation of 2 distinct tumors and corresponding organoids	[107]

Table 2 Selected studies dealing with feasibility of therapy response prediction and selection of patients´ optimal therapy based on PDO drug testing, i.a. [58,91].

The organoid culture system can also be valuable for assessing the therapeutic response of recurrent tumors and potential resistances developing from primary to recurrent tumors. This has been shown by comparing drug responses in matched organoid lines of a high-grade serous (HGS) ovarian cancer patient derived from a primary chemosensitive and recurrent chemoresistant tumor. The increased resistance to platinum-based chemotherapy of the recurrent tumor could be recapitulated in the organoid line established from this tumor [108].

These studies support the feasibility of using PDOs to predict therapeutic efficiency, but the number of patients investigated needs to be increased and most of the studies so far had observational designs. In line with this, an increasing number of trials is listed on the webpage of ClinicalTrials.gov. Another group reporting 30 studies related to cancer organoids was registered in November 2019 [109]. In the meantime, further studies focusing on tumor organoids as a response prediction and treatment guiding model have been added for different tumor entities such as (advanced) pancreatic cancer (NCT04931381, NCT04931394, NCT04736043), ovarian cancer (NCT04555473 TAILOR trial), lung cancer (NCT04859166) or different entities of refractory solid tumors (NCT04279509 SCORE trial) [102]. A relevant percentage are designed as interventional studies such as the SCORE trial (NCT04279509) [110].

These findings showing that patient-derived tumor organoids can predict the tumor responses of individual patients to therapy led to the idea that they can be used to more effectively screen for new individual, therapeutic options than in 2D cell culture systems and can help to identify potential biomarkers for therapeutic efficiency.

Correspondingly, in 2015, van de Wetering et al. set up an organoid library of 20 colon cancer patients, thereby generating 22 tumor organoids and 19 normal adjacent organoid cultures from surgically resected tissue [64]. After characterization of established organoids by conducting whole exome sequencing and RNA expression analysis, they performed a drug screening comprising 83 compounds (25 compounds in clinical use, 10 different chemotherapeutics, 29 compounds tested in clinical trials and 29 experimental compounds). A correlation of the drug sensitivity of different organoids to particular genomic features was performed. As a proof of principle for the feasibility and benefits of organoid-based drug screens, they were able to demonstrate resistance to nutlin-3a, an MDM2 inhibitor, in organoids with TP53 loss-of-function mutation as well as resistance to anti-EGFR inhibitors in a KRAS-mutated background [64]. In a subsequent study, Verissimo et al. employed the biobank of CRC and wildtype organoids from Wetering et al. to further evaluate targeting of the EGRF-RAS-ERK axis for single and combinatory therapeutic approaches in RAS wildtype and mutated backgrounds [111]. They found a beneficial effect of navitoclax, a BCL2/BCLXL inhibitor, in combination with mono or dual inhibition of EGFR-RAS-ERK-pathway (using afatinib or/and selumetinib) on organoid viability in a RAS-mutated setting [111].

In addition to CRC, organoid biobanks have also been established for other tumor entities to screen for and identify potential new therapeutic options and to estimate efficiency of current therapeutics for single individuals. Kopper et al. generated 56 ovarian organoid lines derived from 32 different patients representing pre-malignant as well as malignant neoplasms of the ovary [108]. They found that organoids reliably recapitulate the original tissue at the genome level and reproduce intra-patient tumor heterogeneity. After characterization of the organoids, they performed a drug screen with a library comprising taxane and platinum-based drugs (standard of care in ovarian cancer) as well as more targeted agents addressing the PI3K/AKT/mTOR-, PARP- or Wee1 pathway and gemcitabine. Some of the findings were recapitulated in vivo by setting up a mouse model transplanting some of the generated organoids subcutaneously. They were able to reproduce known clinical responses, for example, of HGS and non-HGS tumors (organoids) to platinum-based chemotherapy and demonstrated the variable drug response of individual organoid lines, which hints towards the complexity of choosing the right treatment regime [108].

Saito et al. established cancer organoid lines from biliary tract cancer (BTC) and were able to culture them in >1 year. On the one hand, they could show that drug sensitivity is associated with gene mutations and gene expression profiles in these BTC organoids, for example, in the context of anti-EGFR inhibitors such as erlotinib. On the other hand, they screened a library of clinically used drugs (339 compounds) for their ability to suppress BTC organoids [73]. In total, 22 of the 339 agents were successfully screened not only comprising classical anticancer drugs but also non-cancer agents such as antifungal drugs (amorolfine) or drugs used in the context of hyperlipidemia (cerivastatin) and Parkinson disease (talipexole) [73]. Furthermore, another drug screen of 24 agents (FDA approved or preclinical molecularly targeted agents) in PDOs from esophageal adenocarcinoma was described by Li et al. [69]. In addition to screening for new therapeutic options, they also correlated the response of their established PDOs to standard chemotherapeutic regimes to patients´ clinical histories and found a good correlation [69].

Combining (low-dose) single drugs can result in synergism and have more success compared to single treatment. Organoid culture is also used for investigation of potentially beneficial drug combinations and reversal of drug resistance. Ponz-Sarvise et al. could demonstrate that in PDAC organoids, but not in untransformed pancreatic organoids, the phosphorylation levels of ERBB2 and ERBB3 were elevated when inhibiting MEK and AKT as a kind of feedback mechanism possibly responsible for insufficient results while targeting MEK/AKT pathway in a KRAS-mutated setting [112]. From this, they showed that a pan-ERBB inhibitor is beneficial in combination with MEK and AKT inhibition in human PDAC organoids, but not to the same extent with an EGFR inhibitor. These positive effects could be extrapolated in a human organoid orthotopic xenograft mouse model by combining a MEK and ERBB inhibitor with measurable tumor shrinking in a short-term intervention study [112]. Furthermore, Pauli et al. established a tumor organoid biobank from 18 different tumor types from patients with metastatic solid tumors including prostate, bladder, kidney, colon/rectum, brain, pancreas, breast, stomach, esophagus, lung, uterus, liver and adrenal gland tumors and characterized them by whole exome sequencing [113]. After selecting four patients with different tumor entities (uterine carcinosarcoma, endometrial adenocarcinoma and two different stage IV CRC patients with differing mutational landscapes) and their corresponding PDOs, they performed high-throughput single cell sequencing, followed by a combinatory drug screening. The results finally were validated in a PDX model [113].

## 6. Organoids as a Valuable Tool of Predicting and Understanding Therapy-Associated Side Effects

When optimizing cancer therapy for individual patients, it is not only essential to improve the effect of the anticancer drug on tumor tissue, but also to design it in a personalized manner. One of the major problems of current chemotherapeutics are adverse side effects limiting applicability over time and finding the correct dosage. To overcome the insufficient rate of new therapeutic agents going through the stages of the the clinical trial process, an optimized method to determine potential severe side effects in the preclinical stage is desirable. The usage of untransformed organoids established from the tissue of tumor origin makes it possible to test tumor specificity of the therapeutic agent [114,115]. Additionally, it has recently been shown that organoids from tissues mainly affected by adverse effects can be utilized to identify and investigate potential problems in advance. For example, central nervous system (CNS) organoids can help to better assess potential neurotoxicity [116,117,118,119]. In this context, Liu et al. assessed the toxic effect of vincristine on cerebral organoids [119]. They found dose-dependent toxicity on both neurons and astrocytes [119]. Moreover, Schielke et al. showed the potential benefit of using brain organoids to optimize radiation regimes in the setting of CNS tumors [118].

The liver is one of the main organs responsible for drug metabolism and is very often affected by different therapeutic agents. Leite et al. generated a human organoid model to test for hepatocyte-dependent, drug-induced liver fibrosis by measuring fibrotic features such as hepatic stellar cell activation and collagen secretion as well as deposition [120]. Mun et al. could show that the hepatic side effects of distinct agents can be recapitulated in liver organoids [121]. They tested drugs withdrawn from clinical use due to their hepatotoxicity in their hepatic iPSC organoids and observed a relevant toxicity as well as even induction of a distinct phenotype such as hepatic steatosis [121].

Finally, mucosal side effects frequently occur during anticancer treatment. Due to high cellular turnover, patients often suffer from oral mucositis due to chemotherapeutic application. Driehuis et al. recently described a model of wildtype oral mucosa organoids to test implications of chemotherapy and potential options to reduce severe side effects [122]. They investigated the beneficial effects of leucovorin on methotrexate-induced mucositis and established an optimal treatment scheme to limit the side effects on oral mucosa [122].

## 7. Conclusions and Perspectives

This and other reviews have summarized that the 3D culture system of organoids can be established from almost all types of organs and allows improvement of fundamental understanding of disease onset and progression as well as of drug development and response prediction [5,7,8,11,99,123].

In particular, AdSC-derived organoids are widely used due to their relatively simple establishment and close resemblance to patients´ diseases ex vivo. Nevertheless, one of the main current deficiencies of the high number of AdSC-derived organoids used in cancer research is the limitation of epithelial components with the lack of stromal cells, vascularization and components of the immune system (tumor microenvironment, TME). The absence of this key component represents a clear drawback of this model system. Therefore, over recent years, efforts have been made to integrate immune cells and other components of the TME at least partially in AdSC organoid cultures by setting up different culturing systems [124]. As a “classic” submerged Matrigel system only allows for reconstitution of epithelial cell components, there is need of exogenous co-culturing of immune cells [124]. Cattaneo et al. described a method that allowed generation of tumor-reactive CD8^+^ T-cells within two weeks with up to 33–50% efficiency from NSCLC or microsatellite-instable (MSI) CRC by co-culturing tumor organoids with corresponding peripheral blood lymphocytes [125,126]. They demonstrated the ability of these tumor T-cells to kill (corresponding) tumor organoids. In line with this, Tsai et al. published a method of co-culturing PDOs from PDACs with patient-matched cancer-associated fibroblasts that play an important role in PDAC disease even for immune infiltration and peripheral blood lymphocytes [127]. Likewise, Chakrabarti et al. elucidated the role of Hedgehog signaling in tumor cell PDL-1 expression and therefore the prevention of cytotoxic T lymphocytes’ (CTLs) effector function by usage of gastric wildtype and tumor organoids as well as (pulse activated) dendritic and CTLs [128]. Votanopoulos et al. designed melanoma-associated organoids enriched by immune cells from matching lymph nodes and demonstrated that this culture setting is able to predict patients´ performance to immune therapeutics such as nivolumab with 85% accuracy [107].

In addition to the submerged Matrigel culture system, further culture options such as air–liquid interface (ALI) or 3D microfluidic culture systems have been described that allow studies on endogenous tumor–immune interactions while preserving the original TME [31,33,124,129]. Such progress further positions organoid research as a valuable tool to study the interaction of tumors and their microenvironments in a personalized manner as patient´s tumor and immune cells are combined in a single assay. Nevertheless, there is room for improvement in culture conditions and accessibility as well as on reducing workload.

In order to expand organoid systems to almost all organs ex vivo, several issues need to be addressed. In particular, AdSC-derived organoids lack vascularization as they are limited to endoderm-derived cell types. Investigations into the vascularization of organoids are ongoing but standardized procedures remain to be established [130,131]. The heterogeneity in organoid size and shape makes standard functional assays difficult to develop, as does the lack of standardized culture conditions. These factors need to be considered when working with organoids for cancer research and drug development [132].

To conclude, the application of organoids in cancer research is rapidly increasing and they have the potential to be a reliable ex vivo model for cancer drug development.

## Figures and Tables

**Figure 1 cancers-13-04812-f001:**
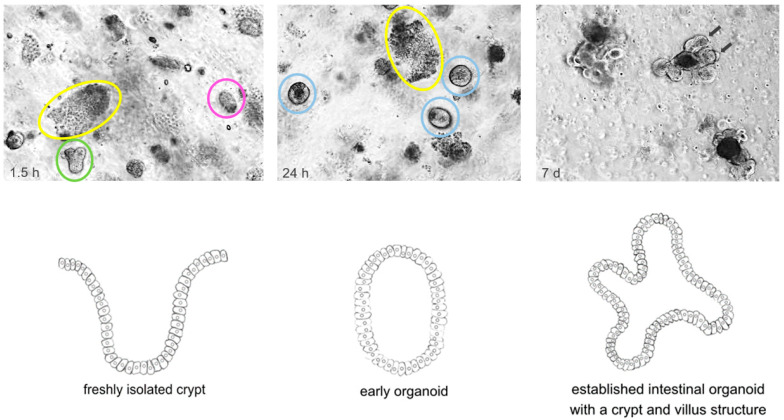
Time course of the isolation and establishment of murine wildtype intestinal organoids. Crypt isolation from surgical resected murine small intestine was carried out using EDTA and mechanical disruption according to initial description of Sato et al. [13]. The upper panel shows light microscopy pictures taken at indicated timepoints. The lower panel graphically illustrates the establishment of a murine intestinal organoid referring to the timepoints of the upper panel. Upper panel: Left: 1.5 h after isolation crypts (green) and villi (yellow) are embedded in Matrigel; some of the crypts have already closed (pink). Middle: 24 h after isolation the upper opening of crypts became sealed and early, round intestinal organoids (light blue) grew out. Villi (yellow) started to degrade. Right: 5–7 days after isolation, fully established intestinal organoids with a crypt (arrows) and villus structure can be found. Lower panel: Left: freshly isolated crypt. Middle: Early organoid structure after upper opening of crypts became sealed. Right: Fully established intestinal organoid with a crypt and villus structure.

**Table 1 cancers-13-04812-t001:** Exemplary summary of published PDO biobanks of different tumor entities.

Tumor type	Sample Size	Type of Specimen	Additional Information	Reference
Colorectal Cancer	22 tumor organoid lines from 20 different colon cancer patients and matched organoids from adjacent untransformed tissue of 19 patients	surgially resected tissue only primary lesions	first organoid biobank described only untreated patients	[64]
	49 organoids from primary lesions (15 premalignant lesions (tubular/tubulovillous/serrated), 32 adenocarcinoma, 2 neuroendocrine carcinoma) and 6 organoid lines from metastatic lesions of adenocarcinomam; additionally 41 counterpart organoids from normal colorectal mucosa	endoscopic biopsy specimen or surgically resected sample primary as well as metastatic lesions	including rare histological subtypes such as poorly differentiated or mucinous adenocarcinoma or neuroendocrine neoplasms	[55]
Gastric Cancer	46 organoid culture lines from tumor or dysplastic lesions of 34 patients and 17 organoid lines from adjacent untransformed mucosa	surgically rescted tissue primary lesion and lymph node metastases	predominantly untreated patients (three patients with neoadjuvant chemotherapy)	[70]
Liver Cancer	10 HCC-derived organoid lines of 8 patients and corresponding normal liver organoids from all of the patients	needle biopsy specimen primary lesions (for 5 patients 2 different nodules were biopsied)	HCC tumors of different etologies (viral hepatitis, NAFLD, ALD) successful establishment in only 26% of patient samples	[72]
	tumor organoids of 3 patients with IHCC and 1 patient with GBC (additionally 1 organoid line from PDAC and 1 organoid line from neuroendocrine tumor of ampulla vateri)	surgically resected tissue	sucessful longterm culture of organoids in 6 out of 18 cases	[73]
Pancreatic cancer	52 (31 analyzed) organoid lines from different subtypes of pancreatic cancer or distal bile duct carcinoma (63% PDAC, 10% CC, 6.67% ACC, 3.33% adenosquamous PDAC, 10% IPMN derived PDAC, 6.67% papilla vateri AC); matched normal coontrol organoids of tumor-adjacent normal tissue from 5 patients	predominantly surgically resected tissue, only 2 biopsy samples		[74]
	49 PDAC organoid lines and normal pancreatic organoids of adjacent untransformed tissue whenever possible to establish	fine-needle aspiration, ascites specimen or surgically resected tissue only primary lesions (or ascites)	tumor stage of all patinets was III or IV except for 2 patients (IIA and IIB) only 3 patients were pre-treted before sampeling	[49]
	114 organoid cultures from 101 PDAC patients; additionally 11 human normal pancreatic ductal organoids from healthy normal pancreata obtained from islet transplant centers	surgically resected tissue, FNB samples or specimen from rapid autopsies primary as well as metastatic lesions		[75]
	10 (8 analyzed) organoid cultures from patients diagnosed with IPMN and 7 additional normal pancreatic duct organoids	surgically resected specimen		[76]
	15 organoid lines from patients with IPMN and normal pancreatic organoids of matched adjacent normal mucosa	surgically resected specimen	comprising 3 low-grade IPMNs, 2 moderate-grade IPMNs, 7 high-grade IPMNs, and 3 IPMNs associated with invasive carcinoma	[77]
Neuroendocrine Tumors	25 organoid lines from pastients with gastroenteropancreatic neuroendocrine neoplasms (NEN)	surgically resected tissue and biopsy samples	comprising NEN from esophagus, stomach, duodenum, colon, liver, pancreas and biliary tract	[78]
Lung Cancer	organoid lines from 10 patients with NSCLC	mainly surgically resected tissue		[79]
	80 organoid lines from patients with different subtypes of lung cancer (66.25% adenocarcinoma, 6.25% small cell lung cancer, 3.75% large cell carcinoma, 3.75% adenosquamous carcinoma, 20% squamous cell carcinoma) and 5 organoid lines from normal bronchial tissue	surgically resected tissue	small number of banked organoid lines from pulmonal metastatic lesions of colonic adenocarcinoma	[80]
	12 orgnoid lines from 15 patients with lung adenocarcinomas of different subtypes	surgically resected tissue		[81]
	organoid lines from 103 surgically resected specimens (71 ACS, 23 SCCs, 4 SCLCs and 5 other lung cancer types); among these 103 specimens 42 pairs of tumor and corresponding normal tissues processed in paralle	surgically resected tissue 3 samples gained by endobronchial ultrasound-guided transbronchial needle aspiration	samples from all tumor stages	[82]
Glioblastoma	70 organoid lines from patients with glioblastoma	surgically resected tissue		[83]
Bladder Cancer	tumor organoid lines from 53 patients with bladder cancer (basal and luminal bladder cancer subtypes); whenever possible corresponding normal organoid lines from untransformed mucosa	surgically resected tissue (cystectomy or transurethral resection (TUR))		[71]
	22 PDO lines from 16 patients ranging from low-grade non-muscle invasive disease to muscle-invasive cancer	surgically resected tissue (TUR)	1 patient was systemically treted before sampeling, 6 organoid lines established from patients with a prior intravesical treatment before sampeling	[84]
Kidney cancer	54 organoid lines from different subtypes of pediatric kidney cancer (40 Wilms tumors, 7 MRTKs, 3 RCCs, 2 CMNs, 1 metanephric adenoma and 1 nephrogenic rest); whenever possible organoid line from corresponding healthy tissue	surgically resected tissue (nephrectomy) or biopsy sample mainly primary lesion, but als metastatic lesions	first pediatric organoid biobank chemo-naïve as well as chemo-treated specimens	[85]
Breast Cancer	>100 breast cancer organoid lines representing distribution of ductal, lobular, adeno- and carcinoma in situ as well as all types of receptor combination (ER, PR, HER2)	surgically resected tissue	chemo-naïve as well as chemo-treated specimens	[63]
	33 organoids from 33 patients with breast cancer; 84.84% invasive ductal carcinoma and 15.15% invasive lobular carcinoma	surgically resected tissue and core biopsy specimens		[86]
	64 organoid lines from patients with triple negative breast cancer			[87]
	17 tumor organoid lines from 32 patients with invasive ductal or lobular carcinoma as well as organoid lines from tumor-adjacent normal tissue	surgically resected tissue	only of treatment-naïve patients	[88]

AC—adenocarcinoma, ALD—alcoholic liver disease, CC—cholangiocellular carcinoma, CMN—congenital mesoblastic nephroma, ER—estrogen receptor, GBC—gallbladder carcinoma, FNB—fine needle biopsy, HCC—hepatocellular carcinoma, HER2—human epidermal growth factor receptor, IHCC—intrahepatic cholangiocellular carcinoma, IPMN—intraductal papillary mucinous neoplasm, MRTK—malignant rhabdoid tumor of the kidney, NAFLD—nonalcoholic fatty liver disease, NSCLC—non-small cell lung cancer, PR—progesterone receptor, RCC—renal cell carcinoma, SCC—squamous cell carcinoma, SCLC—small cell lung cancer, TUR—transurethral resection; adapted and expanded from [89].

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
