# Peer review of "Organoid Models for Cancer Research—From Bed to Bench Side and Back"

_cancers, 2021, doi:10.3390/cancers13194812_

Round 1
Reviewer 1 Report
This is a comprehensive and informative review regarding the use of 3D organoids, especially patient-derived organoids, for cancer research, mainly drug screening. The authors thoroughly reviewed essential references in the field, including pioneering papers from Clevers group, and discussed the utility of different organoids in varieties of approaches of cancer research, which make this review informative to the readers. One aspect this reviewer thinks important with regard to utilization of organoids in cancer research is that organoid system is a useful platform for understanding basic aspect of tumorigenesis, especially from developmental point of view. As an example, Parisian et al (PMID: 32912900) presented that SMARCB1, a tumor suppressor gene in a subset of pediatric brain tumors, is differentially affecting growth of the cells in different developmental states, suggesting that certain cells at a specific differentiation stage may be playing a role in cancer development. It'd make this review even more informative if the authors could discuss the use of these organoids in more basic cancer research as well.
Author Response
We thank the reviewer for his positive feedback. We adopted the manuscript according to the suggestions and included a part about the use of organoids for assessing basic aspects of tumorigenesis.

Reviewer 2 Report
The Review by Kastner et al. describes adult and iPSC derived organoids with a focus on the application of organoids in matching patient responses to therapy and applications in studying signaling pathways, genome editing, and assessment of drug toxicities. While this topic has been covered recently by many other excellent reviews, the Review by Kastner et al. when addressing the concerns below could provide novel and interesting analyses of the use of organoid models to detect responses to therapy.
Major concerns:
1) The authors describe the applications of organoids in predicting responses and drug side effects, which are useful sections, however, the earlier sections to introduce different organoid and/or 3D culture and components are limited and should be expanded to support the data presented in the later sections.
2) The images in Figure 1 are of poor quality. The text is unclear and the timeline needs to be better defined (what does few days mean?).
3) Line 101: the section entitled “Organoids as “Close-to-Reality” Models” describes genome editing studies in mouse CRC models without fulfilling the promise in this section’s title. Further evidence in other organoid types and other applications need to be included to match the claims in the section title.
4) Table 1 is very difficult to follow, it needs to be revised into columns for the different types of data included.
5) The section of using organoids for predicting responses to therapy needs to be with its own heading, as it includes some novel analyses of study findings that could be of interest to the reader.
Minor concerns:
1) The abstract statements regarding organoid novelty and generalization are overstated and discount the historic development of 3D cultures (see Weiswald et al, Neoplasia 2015).
2) Line 79: The statement about organoids being “limited to the epithelial cell type of origin [7,8,24].” And citations excludes other recent evidence of the contrary (e.g., Neal et al., Cell 2018 and others).
4) Language editing is recommended (some examples below):
- Simple Summary: diseased tissue was developed which stably retains the genetic and phenotypic.
- Line 170: a high number of patients.
- Line 407: the application of organoids … and has the potential to be a reliable ex vivo model
- Expressions used PDO in some sections and PTDO in others (e.g., Table 1)?
- There are overall frequent extra spaces between words.
Author Response
Major concerns:
1) The authors describe the applications of organoids in predicting responses and drug side effects, which are useful sections, however, the earlier sections to introduce different organoid and/or 3D culture and components are limited and should be expanded to support the data presented in the later sections.
We agree with the reviewer and have expanded the section to improve introduction in the field.
2) The images in Figure 1 are of poor quality. The text is unclear and the timeline needs to be better defined (what does few days mean?).
The reviewer is absolutely correct, we have to apologies for this. We now edit the format and include more detailed timelines.
3) Line 101: the section entitled “Organoids as “Close-to-Reality” Models” describes genome editing studies in mouse CRC models without fulfilling the promise in this section’s title. Further evidence in other organoid types and other applications need to be included to match the claims in the section title.
We now adopted the title and the content of this part as suggested by the reviewer.
4) Table 1 is very difficult to follow, it needs to be revised into columns for the different types of data included.
We thank the reviewer for this comment. We now completely revised table 1 to be easier to follow.
5) The section of using organoids for predicting responses to therapy needs to be with its own heading, as it includes some novel analyses of study findings that could be of interest to the reader.
This is a good point, we now include this section within it´s own heading.
Minor concerns:
1) The abstract statements regarding organoid novelty and generalization are overstated and discount the historic development of 3D cultures (see Weiswald et al, Neoplasia 2015).
We now turned down this statement and included the suggested reference.
2) Line 79: The statement about organoids being “limited to the epithelial cell type of origin [7,8,24].” And citations excludes other recent evidence of the contrary (e.g., Neal et al., Cell 2018 and others).
Thank you for this reference, we now adopted the text in accordance to your suggggestion.
4) Language editing is recommended (some examples below):
- Simple Summary: diseased tissue was developed which stably retains the genetic and phenotypic.
- Line 170: a high number of patients.
- Line 407: the application of organoids … and has the potential to be a reliable ex vivo model
- Expressions used PDO in some sections and PTDO in others (e.g., Table 1)?
- There are overall frequent extra spaces between words.
We thank the reviewer for this advice. We again had the manuscript been proof reading by a native speaker.
